# Effect of High-Pressure Torsion and Annealing on the Structure, Phase Composition, and Microhardness of the Ti-18Zr-15Nb (at. %) Alloy

**DOI:** 10.3390/ma16041754

**Published:** 2023-02-20

**Authors:** Dmitry Gunderov, Karina Kim, Sofia Gunderova, Anna Churakova, Yuri Lebedev, Ruslan Nafikov, Mikhail Derkach, Konstantin Lukashevich, Vadim Sheremetyev, Sergey Prokoshkin

**Affiliations:** 1Department of Materials Science and Physics of Metals, Ufa University of Science and Technology, Zaki Validi St. 32, 450076 Ufa, Russia; 2Laboratory of Solid State Physics, Institute of Molecule & Crystal Physics, UFRC RAS, 151 Prospect Oktyabrya, 450075 Ufa, Russia; 3Metal Forming Department, National University of Science and Technology “MISiS”, Leninsky Ave. 4, p. 1., 119049 Moscow, Russia

**Keywords:** Ti-18Zr-15Nb shape memory alloys, high-pressure torsion, phase transformations during heating

## Abstract

The Ti-18Zr-15Nb shape memory alloys are a new material for medical implants. The regularities of phase transformations during heating of this alloy in the coarse-grained quenched state and the nanostructured state after high-pressure torsion have been studied. The specimens in quenched state (Q) and HPT state were annealed at 300–550 °C for 0.5, 3, and 12 h. The *α*-phase formation in Ti-18Zr-15Nb alloy occurs by C-shaped kinetics with a pronounced peak near 400–450 °C for Q state and near 350–450 °C for HPT state, and stops or slows down at higher and lower annealing temperatures. The formation of a nanostructured state in the Ti-18Zr-15Nb alloy as a result of HPT suppresses the *β→ω* phase transformation during low-temperature annealing (300–350 °C), but activates the *β→α* phase transformation. In the Q-state the *α*-phase during annealing at 450–500 °C is formed in the form of plates with a length of tens of microns. The *α*-phase formed during annealing of nanostructured specimens has the appearance of nanosized particle-grains of predominantly equiaxed shape, distributed between the nanograins of *β*-phase. The changes in microhardness during annealing of Q-specimens correlate with changes in phase composition during aging.

## 1. Introduction

Titanium (Ti) and its alloys exhibit good biocompatibility, relatively high strength, low density, and high corrosion resistance. In this regard, technically pure titanium and Ti alloys are widely used as a material for orthopedic implants [1]. However, “alpha” and “alpha+beta” Ti alloys have a significantly higher elastic modulus (100–120 GPa) than the elastic modulus of bone (0.1–30 GPa). This dissimilarity leads to development of “stress-shielding effect”, and the bone undergoes resorption of the tissue around the implant [2]. The solution to this problem is the use of alloys with a lower elastic modulus (30–50 GPa), such as metastable “beta” titanium alloys of the Ti-Nb-Zr system [3,4,5,6,7].

These alloys consist of non-toxic components with high corrosion resistance and demonstrate unique biomechanical compatibility with bone tissue. The high biomechanical compatibility of Ti-Nb-Zr alloys is due to the so-called shape memory effect manifesting itself in superelastic behavior—the ability of these alloys to undergo significant (up to 6%) deformation under load and to fully restore their original shape after stress removal [8,9]. This effect is due to the fact that, when the load is applied, the deformation occurs due to the transition of the crystal lattice into an orthorhombic *α*″-martensite. When the load is removed, the reverse transition to the initial BCC *β*-phase occurs. As a result, deformation occurs in the opposite direction and the material restores its original structure and, with it, its shape.

The leader among the shape memory alloys (SMA) in application and in study is titanium nickelide (nitinol)—an intermetallide of close equiatomic composition Ti50:Ni50 [10]. This alloy is the main SMA material for engineering and medical applications. However, it contains Ni, which can be allergenic, which limits the use of Ti-Ni in medicine. In this regard, Ti-18Zr-15Nb (at. %) SMA, which does not contain Ni, is of particular interest [8,9,10]. This alloy in the quenched state (after annealing 700–800 °C following by water quenching) is in the state of single BCC *β*-phase [11]. In this state, Ti-18Zr-15Nb alloy exhibits low elastic modulus (30–50 GPa) and superior superelasticity [12]. It was also shown in [13] that the thermomechanically treated Ti-18Zr-15Nb alloy exhibits the sufficient corrosion resistance and high fatigue life during functional cycling in Hanks’ solution. However, during annealing at 300–500 °C, the beta phase can be unstable and decay with the release of HCP *α*- and FCC *ω*-phases [14]. The presence of these phases in the composition contributes to an increase in strength and deterioration of functional properties (increase in Young’s modulus and degradation in superelasticity) [14].

An additional increase in the service and strength characteristics of Ti-18Zr-15Nb SMAs is an important aim. It can be achieved by formation of a nanostructured state (NS) using methods of severe plastic deformation (SPD) [15] and, in particular, by high-pressure torsion (HPT) and equal-channel angular pressing (ECAP). Previously, these methods have been shown to be effective for forming a nanostructure and for improving the properties of titanium [16] and a number of titanium alloys for medical applications, including Ti-Ni SMAs [17,18,19,20]. Recently, studies on the effect of SPD on Ti-18Zr-15Nb alloys have started [21,22,23]. The ECAP leads to formation of a predominantly nanocrystalline structure of *β*-phase with grains and subgrains of about 100 nm, and to an almost twofold increase in the yield strength [22,23]. The *β*-phase is a main phase in the Ti-18Zr-15Nb after HPT with a certain amount of *α*″-phase, and the grains after HPT processing are refined down to 50 nm [21]. However, many questions remain open. For example, it is of great interest to investigate the *β*-phase stability in specimens pre-treated by HPT. As a result of HPT, the structure of Ti-18Zr-15Nb alloy significantly changes, the nanostructural state is formed, and the kinetics of *α*- and *ω*-phases’ precipitation should noticeably change.

Many studies have been previously performed on the HPT effect on the structure of the *β*-titanium alloys and the commercially pure Ti [24,25,26,27,28,29]. As a result of HPT, a nanocrystalline structure with an *α*-phase grain size of about 100 nm is formed in CP Ti. It is known that HPT accelerates the formation of the *ω*-phase [24,25,26]. The *ω*-phase disappears as a result of the reverse *ω→α* transformation during heating after HPT. It was shown in [27,28,29] that HPT of the beta-titanium Ti-15Mo alloy leads to a decrease in the size of *β*-grains to 80 nm. Grain refinement is accompanied by *β↔ω* phase transformation and non-monotonic change of *ω*-phase volume fraction. The volume fraction of *ω*-phase particles as a result of HPT with one turn of anvils reaches 10% [27,28,29]. With the subsequent increase of HPT degree (number of turns), the volume fraction of *ω*-phase was decreased due to the reverse *ω→β* transformation. It was found that the formation of nanostructure as a result of HPT in Ti-15Mo alloy led to a change in the kinetics of *α*-phase release during subsequent aging, which was characterized by an increase in the nucleation centers of *α*-phase compared to the coarse-grained state and an increase in the volume fraction of *α*-phase (6 and 4 times after aging for 30 min at 500 and 550 °C, respectively) [28]. The formation of *α*-particles in the nanostructured Ti-15Mo is predominantly equiaxial in contrast to the lamellar shape in the coarse-grained alloy.

The kinetics of precipitation hardening during heating in severely deformed alloys of the Ti-Ni system changes in the same complex way [30,31,32]. Ti-Ni alloys, when exposed to HPT, amorphize. Thus, in HPT heated Ti-Ni alloys with excess of Ni, the release of Ti_3_Ni_4_ phase aging particles occurs simultaneously with the crystallization of the amorphous phase. This complicates the analysis of precipitation hardening in HPT-treated Ti-Ni alloys.

In order to obtain the required semi-product, Ti-18Zr-15Nb SMA (including nanocrystalline state) can be processed at elevated temperatures, which can lead to phase transformations. It is known that the appearance of *α*- and *ω*-phases in Ti-18Zr-15Nb affects their functional properties (Young’s modulus and superelastic behavior). There is an obvious question of *β*-phase stability at different temperatures in HPT-treated alloy Ti-18Zr-15Nb. Thus, the purpose of this work is to study the stability of the *β*-phase during annealing under different modes of Ti-18Zr-15Nb alloy in the initial quenched state and after preliminary exposure to HPT.

## 2. Materials and Methods

A 20 kg-weight, 160 mm-diameter ingot with a nominal atomic composition Ti-18Zr-15Nb (at. %) was fabricated by vacuum arc melting and then was subjected to multi-axial hot forging in the 950–1050 °C temperature range to obtain bars of 20 mm diameter. To eliminate the “prehistory” in the alloy, the bars were subjected to annealing at 700 °C (30 min) followed by quenching in water (Q-specimens). At this annealing temperature, recrystallization of the alloy occurs in the specimen, and *α*-, *α*″-, and *ω*-phases are completely transformed into the main *β*-phase. For further research, including XRD investigations, metallography, HV measurement, and HPT treatment, discs with a thickness of 1 mm and a diameter of 20 mm, respectively, were cut from the bars using electrical discharge machining (EDM). A specimen in the form of a thin disc with a thickness of about 1 mm and a diameter of 20 mm was used for high-pressure torsion. The Ti-18Zr-15Nb nanostructured specimens (HPT-specimens) were obtained by HPT at room temperature and 6 GPa pressure on 20 mm diameter anvils with a groove of 0.5 mm depth. The number of anvil rotations was n = 5. The specimens were annealed at 300–550 °C for 0.5, 3, and 12 h in a Nabertherm B180 muffle furnace. After annealing, the surface layer of the material was removed on a grinding machine, and the surface was polished.

The phase composition was analyzed by XRD using an Advance D8 diffractometer (Bruker, MA, USA) at room temperature. Cu-Kα radiation was used and the analysis was conducted in the range of Bragg angles 2θ from 20° to 130°. The volume fraction of the *α*-phase $V_{\alpha }$ was calculated as:(1)$$V_{\alpha }=\bar{I_{hkl}^{\alpha }}/\left(\bar{I_{hkl}^{\alpha }}+\bar{I_{hkl}^{\beta }}\right)$$

The relative integrated intensity of the X-ray lines is a function of several factors, as follows [33].
(2)$$I_{hkl}=\frac{I_{0}}{LP\left(\theta_{hkl}\right)\times P_{hkl}\times {\left|F\right|}_{hkl}^{2}\times e^{-2M}}$$

The primary intensity $I_{0}$ was determined from the obtained diffractograms according to relation (3), where $I_{peak}$ is the peak intensity, $I_{back}$ is the background intensity, and $B_{hkl}$ is the half-width of the diffraction lines.
(3)$$I_{0}=\frac{\left(I_{peak}-I_{back}\right)\times B_{hkl}}{I_{back}}$$

Lorentz-polarization factor $LP\left(\theta_{hkl}\right)$ is defined as:(4)$$LP\left(\theta_{hkl}\right)=\left(1+\cos^{2}2\theta \right)/\left(\sin^{2}\theta \times \cos\theta \right)$$

Multiplicity factor $P_{hkl}$ is given in Table 1 as a function of the Miller indices of the planes and the crystal structure. Structure factor ${\left|F\right|}_{hkl}^{2}$ depends on the atomic scattering factor *f* of the elements in the phase and the crystal structure of the phase, and is given in Table 1.

Atomic scattering factor *f* is a function of $\sin\theta_{hkl}/\lambda$; the most accurate values of these values are given in the tables [34]. The weighted average of multiple components is used for *f*:(5)$$f=f\left(\frac{\lambda }{2\cdot \sin\theta_{hkl}}\right)=f_{Ti}\times 0.67+f_{Zr}\times 0.18+f_{Nb}\times 0.15$$

The value *M* a part of the temperature factor $e^{-2M}$ is determined as:(6)$$M=\frac{6h^{2}}{m\times k\times \theta_{D}}\times \left(\frac{1}{4}+\frac{\Phi \left(\frac{\theta_{D}}{T}\right)}{\left(\frac{\theta_{D}}{T}\right)}\right)\times \frac{\sin^{2}\theta }{\lambda^{2}}$$
where *h*—Planck’s constant, *m*—the mass of the vibrating atom, *k*—Boltzmann’s constant, $\theta_{D}$—the Debye characteristic temperature of the substance in °K, and $\Phi \left(x\right)$—is a function tabulated, along with values of $\theta_{D}$.

The microhardness of specimens in different states was measured using the device DuraScan 50 (according to GOST 2999-75 Vickers hardness), with a load of 200 g for 10 s.

The microstructure and substructure of the specimens were studied by transmission electron microscopy (TEM) using a JEOL 2100. Thin foils for TEM study were obtained by cutting a 3 mm diameter disk from a 100 µm thinned plate. The obtained disk was subjected to electropolishing on a Tenupol-5 unit in an electrolyte: 6% perchloric acid, 35% butanol, 59% methanol with a voltage of 20 V, and temperature—20 °C.

Metallographic studies of the microstructure were carried out using an optical microscope “OLYMPUS GX51”. The investigated surface was mechanically ground and polished on diamond pastes with a dispersion of abrasive from 7/5 to 3/2 microns. To reveal the microstructure, an etchant of the following composition was used: 60%H_2_O + 35%HNO_3_ + 5%HF.

## 3. Results

### 3.1. XRD Results

A single-phase structure consisting of *β*-phase grains is formed in Ti-18Zr-15Nb alloy (Q-state) after quenching from 700 °C. Grain size in the quenched state is about 10 microns, as previously observed for similar states of alloy Ti-18Zr-15Nb [21]. The HPT in n = 5 turns at room temperature leads to the formation of a nanostructured state with the *β*-phase grain/subgrain size less than 50 nm and a high dislocation density [14]. Thus, structural states of the alloy after quenching and after HPT are fundamentally different and this should have a significant effect on the phase transformations during subsequent heating.

Results of the XRD studies of the Ti-18Zr-15Nb alloy after quenching and aging for 3 h at temperatures of 300–550 °C are presented in Figure 1. Table 2 shows the results of the XRD-based phase analysis of the Q-alloy Ti-18Zr-15Nb after aging for 0.5 h, 3 h, and 12 h at temperatures of 300 to 550 °C. As expected, *β*-phase is the main phase in the alloy immediately after quenching, while *α*-phase in the alloy is not detected (Table 2). The annealing of Q specimens at 300 °C and 350 °C does not lead to noticeable *α*-phase formation, whereas the *ω*-phase appears in an appreciable quantity after annealing at 300 and 350 °C. It should be noted that the accuracy in determining the *ω*-phase fraction (and in some cases also *α*-phase) is not high. However, the *ω*-phase peaks observable in the diffractograms (Figure 1) allow us to trace the general tendency of the phase composition change. The *ω*-phase is not registered after annealing at 300 °C 0.5 h. When the annealing time is increased to 3 or 12 h at 300 °C, as well as at 350 °C, the *ω*-phase fraction becomes noticeable and is registered by XRD (up to 30%).

The *α*-phase extensively forms at annealing temperatures of 400–450 °C. It can be noted that the *α*-phase fraction after annealing at 400 °C for 0.5 h is small (less than 2%), and when the annealing time is increased to 3 h at 400 °C, its fraction increases to 20% and continues to grow when the annealing time is increased to 12 h. After annealing at 450 °C for 0.5 h, the *α*-phase already appears in a volume fraction up to 25%. Then, a certain increase in its fraction with increasing annealing time up to 3 h is observed, but significant growth of the *α*-phase fraction with increasing annealing time at 450 °C to 12 h does not occur (Table 2). The intensity of *α*-phase formation at 500 °C decreases (Table 2). Only a small fraction of *α*-phase is observed after annealing at 550 °C.

It is known that a certain texture is formed in the bars after TMT, and the texture can introduce an error in determining the fraction of phases from XRD data. The XRD study of Q-specimens was performed on disc specimens cut from the bar in the transverse direction. To determine the error that can be given by the texture of the bars after TMT in determining the phase fraction, some specimens were cut in the longitudinal direction of the bar and aged at 400 °C for 3 h and 500 °C for 3 h.

The XRD studies showed that the *α*-phase fractions of the specimens cut and “crossed” in the longitudinal direction were determined to be 20 and 23% for the Q + 400 °C 3 h state, and 13 and 11% for the Q + 500 °C 3 h state (Table 2). Accordingly, the texture in the bar after the TMT apparently does introduce error in the determination of the *α*-phase fraction based on XRD data, but this contribution is relatively small and noticeably less than the total measurement error.

The results of the XRD studies of the Ti-18Zr-15Nb alloy after HPT and aging are presented in Figure 2, Table 2. It should be noted that the main sources of large errors in the phase fraction analysis for the HPT specimens are as follows: (1) the effect of the crystallographic texture formed in the HPT specimens is reduced but not completely eliminated by the use of a set of X-ray lines; (2) there is low accuracy in determining the integrated intensities of weak X-ray lines. However, the analysis of phase volume fraction from XRD data allows an understanding of the general picture of phase transformations in the HPT specimens. The *β*-phase is a main phase in the alloy immediately after HPT. The *α*″-phase is a secondary phase which represents itself a stress-induced martensite. Formation of stress-induced *α* and *ω*-phases is doubtful. The regularities of phase transformations during 300–350 °C annealing in the state after HPT are markedly different from those in the Q state. In contrast to Q state, *ω*-phase after annealing at 300 °C and 350 °C of HPT state is not fixed by XRD, probably due to too small *ω* particle size and overlapping of its X-ray diffraction profiles with those of other phases.

However, during annealing, the *α*-phase is actively formed. Its content reaches more than 20% and increases up to 40% during annealing at 350 °C for 12 h (Table 2). Note that the strain-induced *α*″-martensite stabilized by plastic deformation during HPT is preserved during aging up to 300 °C, 0.5 h (Figure 2).

As well as in case of Q state, the *α*-phase formation in HPT specimens is most intensive at annealing temperatures of 400 and 450 °C (Table 2, Figure 3). The *α*-phase content in HPT specimens after annealing for 0.5 h is noticeably higher than in Q specimens. Thus, in Q + 400 °C 0.5 h state, the *α*-phase fraction is negligibly small, while in HPT 400 °C 0.5 h state, the *α*-phase fraction amounts to about 30%. After holding for 12 h at 400 °C, the *α*-phase content in HPT specimens reaches up to 45%. The contents of *α*-phase in Q and HPT specimens with an increase of annealing time up to 12 h at 400 and 450 °C become closer. No *α*-phase is observed after annealing of HPT specimens at 550 °C.

### 3.2. Microhardness Measurement Results

These changes of microhardness during annealing of Q specimens correlate with changes in phase composition during aging (Figure 4a). In the quenched state, the microhardness increases intensively as a result of aging at 300 and 350 °C (Figure 4), which can occur due to the appearance of *ω*-phase. The highest HV values in Q state are reached after of aging at 350 °C 12 h, which can probably be explained by the release of a large fraction of the finely dispersed *ω*-phase. Some growth of HV relative to values in the Q state also occurs during annealing at 400 °C as well as 450 °C (Figure 4). The microhardness increases with increasing holding time at these temperature, but the achieved HV values are less than those at 350 °C. The increase of microhardness after annealing at 400 and 450 °C is caused by the release of an increasing fraction of the *α*-phase. It is known that microhardness/elastic modulus of phases in Ti–alloys are increased in the order of *β→α→ω*-phases. Respectively, *ω*-phase is the hardest of these phases; in addition, *ω*-phase is released in the most finely dispersed form; respectively, its release at 350 °C leads to the greatest increase in HV. The *α*-phase is formed in the form of coarser plates (see the results of TEM below), and its hardness is somewhat lower than that of *ω*-phase. Correspondingly, the formation of even a considerable part of *α*-phase during annealing at 400 and 450 °C results in a lower growth of HV than formation of *ω*-phase at 350 °C.

The microhardness of the alloy upon annealing at 500 °C slightly increases relative to the Q–state (Figure 4), whereas annealing at 550 °C has little or no effect on microhardness because the fraction of *α*-phase is negligibly small and is released at the last temperature.

The HPT treatment results in an almost twofold increase in microhardness (from 200 to 334 HV) compared to the Q state (Figure 4b), which is the result of grain refinement and formation of the nanocrystaline structure. In the HPT state, microhardness increases as a result of aging at temperatures of 300 to 450 °C (Figure 4), due to the appearance of finely dispersed *α* -phase. However, the relative growth of microhardness in HPT specimens after annealing at 300 and 350 °C is appreciably lower than in the Q specimen. This is possibly due to a larger *ω*-phase amount forming in the Q state at these temperatures. HV of the Q + 12 h 350 °C and HPT + 12 h 350 °C states become close, although the mechanisms of hardening for these states are not the same. There are formations of a significant volume fraction of finely dispersed *ω*-phase in the state Q + 12 h 350 °C and nanostructured state with a formation of finely dispersed *α*-phase in HPT + 12 h 350 °C state.

The HV maximum (≈410) at 400 °C HPT state is reached at 3 h (Figure 4); at 450 °C the HV maximum is reached at 0.5 h (also ≈410). With further increase in holding time at these temperatures, the microhardness slightly decreases relative to the maximum as a result of growth of *α*-phase particles and *β*-phase grains. However, as a whole, HV of HPT state after annealing at 400 °C and 450 °C (350–400 units) is appreciably above the microhardness of a Q state after similar annealing (Figure 4), which is the result of preservation of the nanocrystalline structure and fine dispersion of *α*-phases crystals in HPT state after such annealing. Annealing at 500 and 550 °C of the HPT state leads to a decrease in HV relative to the HPT state, which is the result of grain growth, *α*-phase particles growth, and a decrease in its fraction. As a result of 550 °C annealing, the microhardness of the HPT and Q states becomes close to the HV of the original Q state (Figure 4).

### 3.3. TEM and OM Results

According to TEM, *α*-phase plates with a thickness of about 50–100 nm and a length of about 500 nm are formed after annealing at 450 °C 3 h in the Q state (Figure 5a). These plates are observed in both bright-field and dark-field TEM images, the latter are obtained from the reflexes of {110} family (see DF2) and {103}α + {211}β family (see DF1). The *α*-phase plates are observed both in the body of *β*-grains and along the boundaries of the original *β*-grains. Hence, the boundaries of *β*-grains are regions of *α*-phase nucleation. The SAED pattern corresponds to the coarse-grained state of the *β*-phase. The *α*″-phase reflexes are also present in the SAED pattern. It may be noted that the *α*″-phase peaks are not observed in the XRD patterns (see XRD results). The α″-phase X-ray lines «sink» in the X-ray diffractogram background due to their low amount. On the contrary, even weak α″-phase reflexes can be distinctly visible due to their azimuthal and radial resolution. It is consequently possible that the *α*″-phase was formed during preparation of thin foils for TEM, as cooling or/and deformation-induced products.

The microstructure of the Q alloy after increasing the annealing time to 12 h at 450 °C is generally similar to the Q + annealing 450 °C 3 h state. The dark-field images show the presence of *β*-, *α*-, and *α*″-phases, as DF1 is taken from the reflex of {110}*α* family and DF2 is taken from {110}*β* +{100}*α* +{110}*α*″ families.

The nanocrystalline structure of the HPT specimen after annealing at 450 °C for 3 h is observed (Figure 6a), with average in the 30–50 nm range. The reflexes of *β*-phase and *α*-phase form dotted rings in the SAED pattern (Figure 6a), which also indicates the formation of the nanocrystalline structure. The *α*-phase after HPT and annealing at 450° for 3 h are observed as separate equiaxed grain-particles mixed with the equiaxed *β*-phase nanograins (Figure 6a, dark-field images in a DF2 reflex). As in the case of the Q state after such annealing, the *α*″-phase reflexes are observed in the SAED pattern HPT + 450 °C state. As the critical average grain size for cooling-induced *α*″-martensite formation is about 250 nm and that for stress-induced *α*″-martensite formation is 36 nm in the Ti-18Zr-14Nb alloy [21], the observed *α*″-phase should be a stress-induced martensite formed during preparation of thin-foils.

The structure of the HPT state after annealing at 450 °C for 12 h (Figure 6b) is generally similar to that obtained by 3 h annealing. The average grain size somewhat increases; however, it does not exceed 100 nm. In the SAED pattern, both individual bright blurred *β*-phase reflexes and rings of small *β*- and *α*-phase reflexes are observed. The population of reflexes in dotted rings is less dense than after the HPT + 450 °C 3 h, which correlates well with the growth of *β*- and *α*-phase nanosized grains.

Figure 7a,c shows optical microscopy images with different magnifications of the Q alloy microstructure after annealing at 500 °C for 12 h. The *β*-phase grains are observed as in the Q state. Their size is 50–100 μm. The *α*-phase plates, about 5 μm thick and up to 50 μm long crossing the *β*-phase grains, are observed as well (Figure 7a,c). Dark *α*-phase inclusions at the boundaries of *β*-phase grains are also clearly visible (Figure 7c). Thus, the *α*-phase is actively released along the boundaries of *β*-phase grains during annealing, which is consistent with data from other papers [14].

Figure 7b,d shows (OM) images of the microstructure of the alloy after HPT and annealing at 500 °C for 12 h. The grain size of the *β*-phase and *α*-phase is apparently of about 1 µm. This size is at the resolution limit of optical metallography. Thus, the grain size increases compared to the HPT state after 450 °C annealing, but there is no intensive grain growth at 500 °C annealing. The observed bands (Figure 7d) in the structure are apparently strain bands formed during HPT, the contrast of which has increased due to the isolation of the *α*-phase.

After annealing at 550 °C, the microstructure of both the Q and HPT states changed significantly (Figure 7e,f). The *α*-phase plates are practically not observed in the OM images (Figure 7e). The *β*-phase grain size is 20–40 μm while the grain boundaries have less contrast than in the state Q + 500 °C 12 h, since the grain boundaries due to the release of the *α*-phase in the latter state are more contrasting.

Figure 7f shows OM image of the microstructure after HPT and annealing at 550 °C for 12 h. This annealing results in a structure with a *β*-phase grain size of about 10 μm, i.e., recrystallization with intensive grain growth occurred. The *α*-phase in OM image of HPT + 550 °C 12 h state is not identified. Apparently, the intensive grain growth in the HPT state during annealing at 550 °C is explained by the fact that at 550 °C there is no *α*-phase extraction at the *β*-grain boundaries. At the same time, *α*-phase at the *β*-grain boundaries during the annealing at 500 °C in the HPT state actively forms, which largely restrains the growth of *β*-grains.

## 4. Discussion

Thus, the results of XRD study revealed the following pattern of phase transformations during annealing of Q-alloy. The isothermal ω_iso_ phase in Q-alloy during annealing at 300 and 350 °C is formed, as was observed for other Ti *β*-alloys [35]. As the annealing temperature increases, the *α*-phase is formed. A 0.5 h annealing at the lowest (threshold) temperatures for *ω*- and *α*-phase formation (300 °C for *ω*-phase and 400 °C for *α*-phase) does not lead to their active release and they are not registered by XRD. An increase in annealing temperatures of 50 °C (350 °C for *ω*-phase and 450 °C for *α*-phase) at 0.5 h leads to their formation in a quantity of more than 20%. The maximum amount of *α*-phase in Q-state–of about 40% is reached after annealing at 400 °C 12 h. During annealing at 550 °C for 12 h, *α*-phase practically is not formed. The obtained results correlate with common knowledge [25] that the *α*-phase formation in Ti-18Zr-15Nb alloy occurs by C-shaped kinetics with a pronounced peak near 400–450 °C and stops or slows down at higher and lower annealing temperatures (Figure 8).

The formation of the nanostructured state in the Ti-18Zr-15Nb alloy as a result of HPT presumably suppresses the *β → ω* phase transformation during low-temperature annealing but activates the *β → α* phase transformation. The area of *α*-phase formation bounded by the C-curve extends to lower temperatures which correspond to the nanocristalline state (Figure 8). The high density of defects and grain boundaries is characteristic of the nanocrystalline state formed as a result of HPT and HPT, followed by a low-temperature annealing. This apparently leads to a decrease in the temperature of *α*-phase formation since the defects and grain boundaries are the preferable regions of its nucleation.

The absence of reliable evidence of *ω*-phase in diffractograms of the Ti-18Zr-15Nb alloy, both directly after HPT and after HPT and low-temperature annealing, should be discussed further. As already mentioned, in pure titanium and a number of Ti alloys, the HPT at room temperature initiates the formation of *ω*-phase [27,28,29,36]. It has been previously shown that in the initial stages of the HPT (n = 1) in the *β*-Ti alloys Ti-Nb-Ta-Zr, Ti-15Mo [27,28], the *β → ω* transformation occurs, which is caused by the application of high pressures and the formation of a certain concentration of the lattice defects. However, the *ω*-phase could not be identified when the degree of HPT deformation was increased (up to n = 5 and n = 10 turns) [27,36]. It is assumed that the reverse *β → ω* transformation occurs in such alloys upon grain refinement to a certain critical size (less than 100 nm). Thus, the absence of signs of the *ω*-phase in the alloy Ti-18Zr-15Nb (according to XRD data) immediately after the HPT may be due to a number of reasons: (1) complete blocking of the athermal *ω_ath_*-phase formation under HPT due to the specific features of the alloy composition; (2) hyperfine-grained structure of the forming *ω_ath_*-phase, high internal stresses and high density of lattice defects in the initially severely deformed specimens, which did not allow the register of *ω*-phase by XRD method; (3) an “overcritical” degree of HPT deformation resulted in the reverse transformation *ω→β.* However, the explanation “3” apparently is not correct, because in the previous study the *ω*-phase was not registered by XRD method in Ti-18Zr-15Nb alloy immediately after HPT in n = 1 turns [21].

Preliminary HPT also changes the morphology of the *α*-phase, which is formed during the subsequent annealing. In the Q-state, the *α*-phase during the subsequent annealing at 450–500 °C is formed in the form of plates with a length of tens and hundreds of microns. The *α*-phase formed during annealing of nanostructured specimens has an appearance of nanosized particle-grains of predominantly equiaxed shape distributed between nanograins of *β*-phase. Previously, a similar change in the morphology of the *α*-phase formed during the HPT and annealing of *β*-Ti alloy specimens was noted in [27].

## 5. Conclusions

The regularities of phase transformations during heating of the Ti-18Zr-15Nb alloy in the coarse-grained quenched state and the nanostructured state after HPT have been studied.

The formation of the nanostructured state in the Ti-18Zr-15Nb alloy as a result of HPT suppresses the *β→ω*-phase transformation during low-temperature annealing (300–350 C) but activates the *β→α*-phase transformation. In the HPT specimens at 400 °C and small holding times (0.5 h), the *α*-phase evolution rate is noticeably higher than in Q specimens. As the annealing time at 400 °C or temperature (up to 450 °C) increases, the amount of released *α*-phase in Q and HPT specimens converges. The *α*-phase formation in Ti-18Zr-15Nb alloy occurs by C-shaped kinetics with pronounced peak near 400–450 °C for Q state and near 350–450 °C for HPT state, and stops or slows down at higher and lower annealing temperatures.

Preliminary HPT deformation changes the morphology of the *α*-phase, which is formed during the subsequent annealing. In the Q-state the *α*-phase during annealing at 450–500 °C is formed in the form of plates with a length of tens of microns. The *α*-phase formed during annealing of nanostructured specimens has the appearance of nanosized particle-grains of predominantly equiaxed shape, distributed between the nanograins of *β*-phase.

The changes in microhardness during annealing of Q-specimens correlate with changes in phase composition during aging.

## Figures and Tables

**Figure 1 materials-16-01754-f001:**
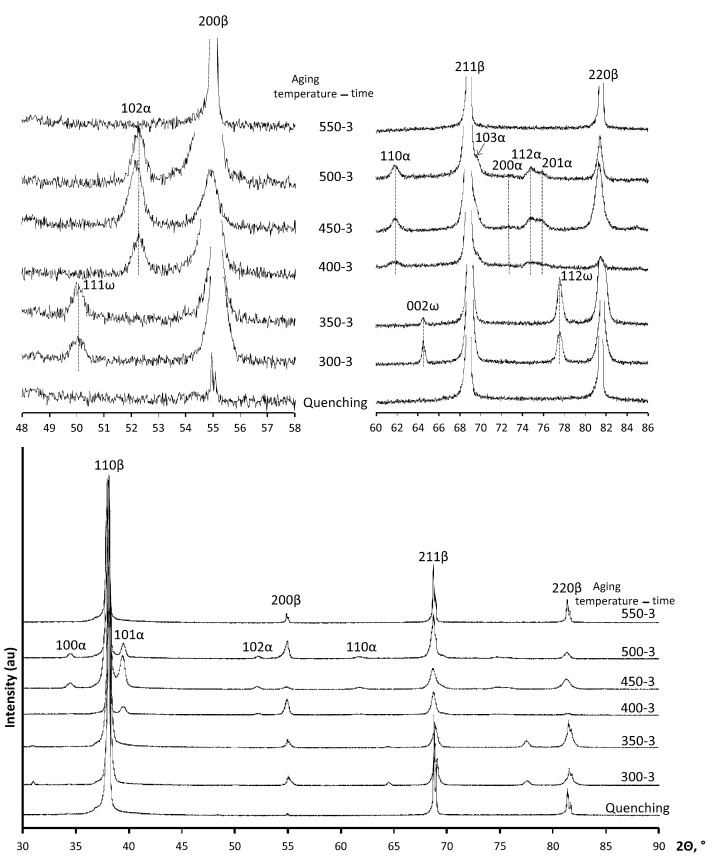
X-ray diffractograms analysis of Ti-18Zr-15Nb alloy after quenching and quenching followed by aging.

**Figure 2 materials-16-01754-f002:**
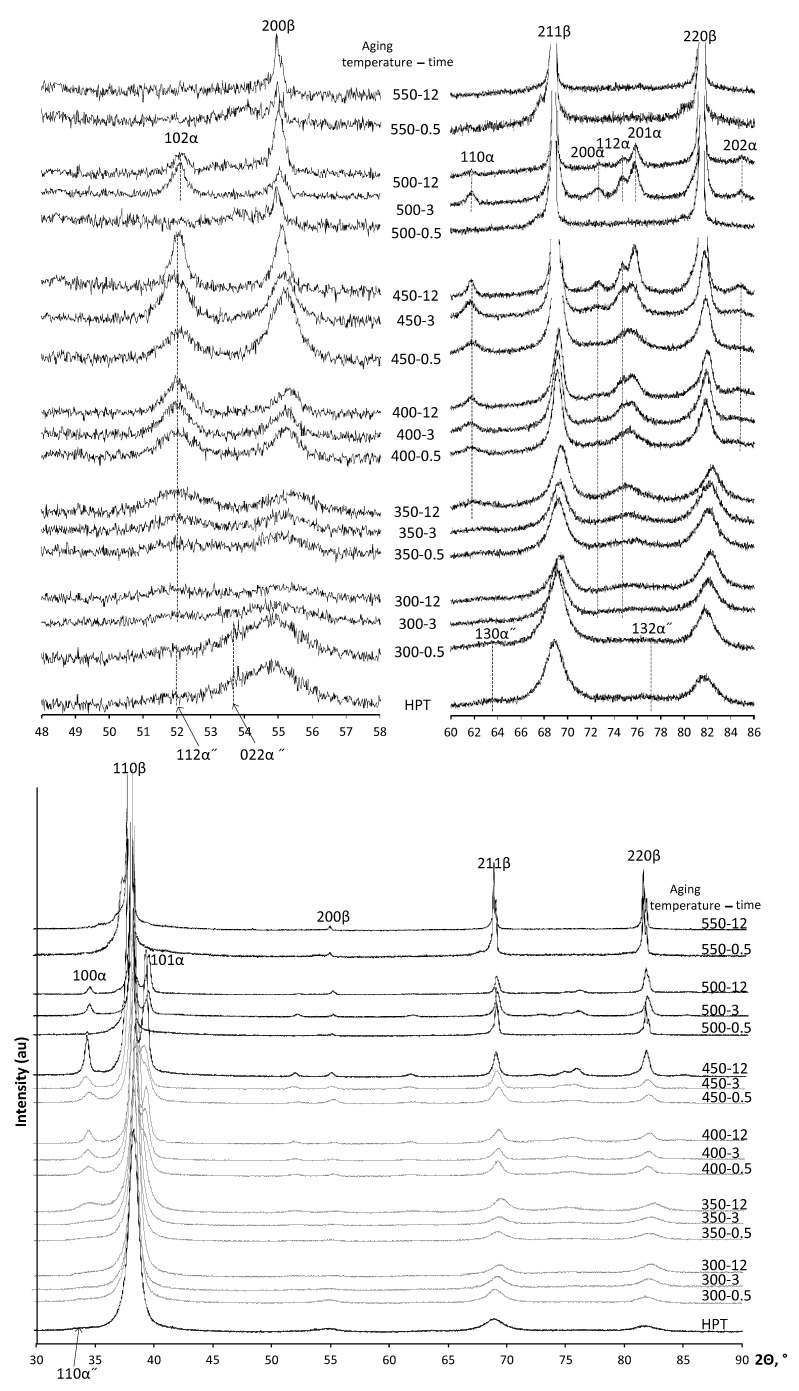
X-ray diffractograms of Ti-18Zr-15Nb alloy after HPT and aging.

**Figure 3 materials-16-01754-f003:**
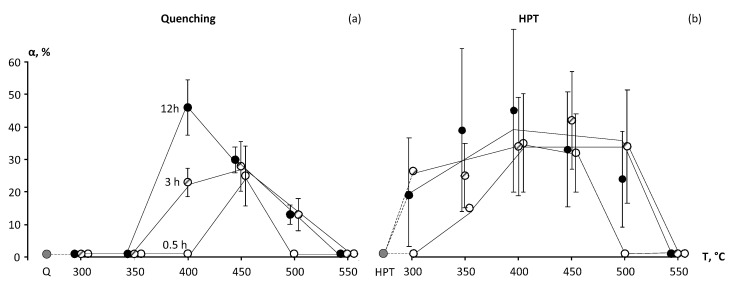
Volume fraction of *α*-phase in the Ti-18Zr-15Nb alloy depending on annealing route after (**a**) initial quenching; (**b**) initial HPT.

**Figure 4 materials-16-01754-f004:**
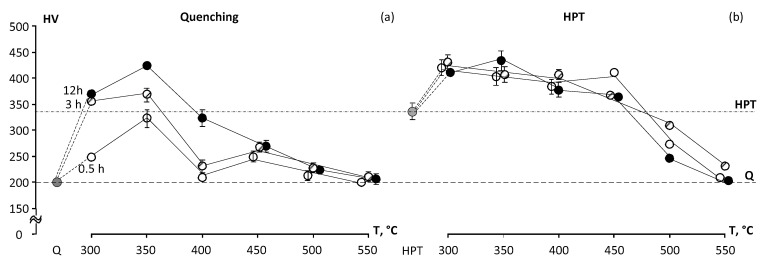
Hardness of the Ti-18Zr-15Nb alloy depending on annealing route after: (**a**) quenching and (**b**) HPT.

**Figure 5 materials-16-01754-f005:**
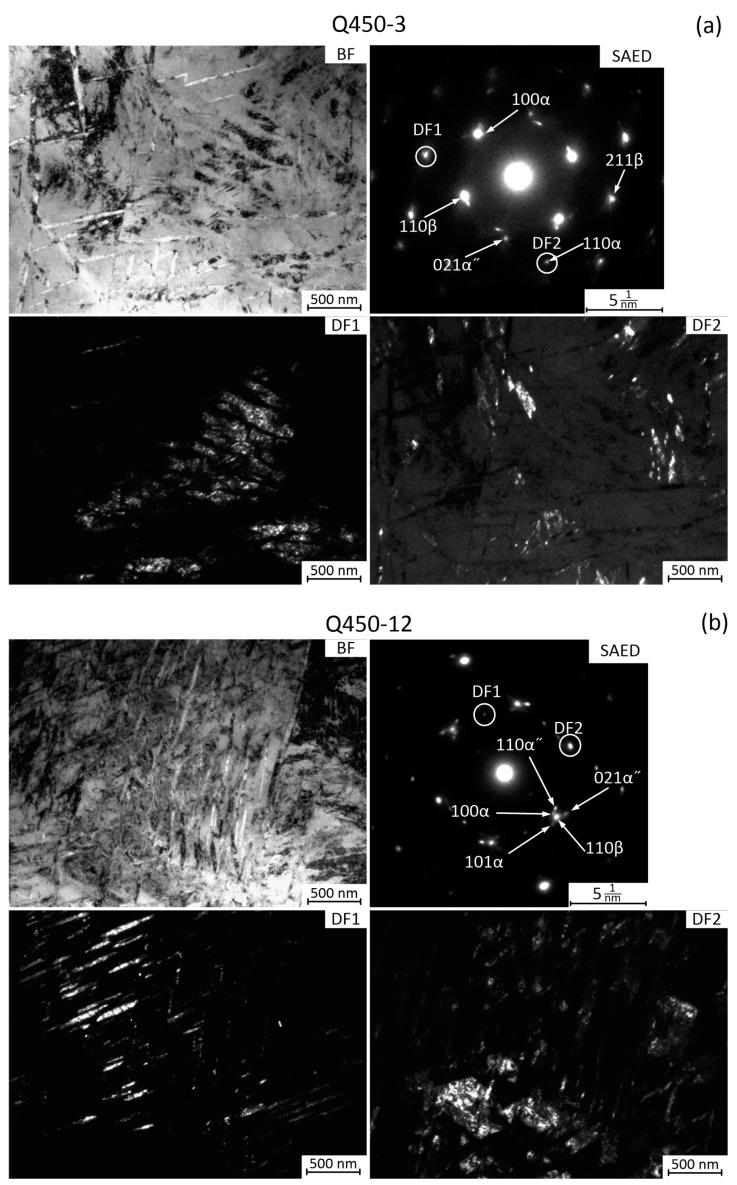
TEM images of the Ti-18Zr-15Nb alloy after: (**a**) Q + 450 °C 3 h; (**b**) Q + 450 °C 12 h. Bright field (BF) and dark field (DF) images, and SAED patterns are shown. Zone axis is close to <110> *β.* In (**a**), DF1 is taken from an individual 211*β* family reflex, while DF2 is taken from an individual 110*α* reflex. In (**b**), DF1 is taken from an individual 100*α* family reflex, while DF2 is taken from 110*β* reflex.

**Figure 6 materials-16-01754-f006:**
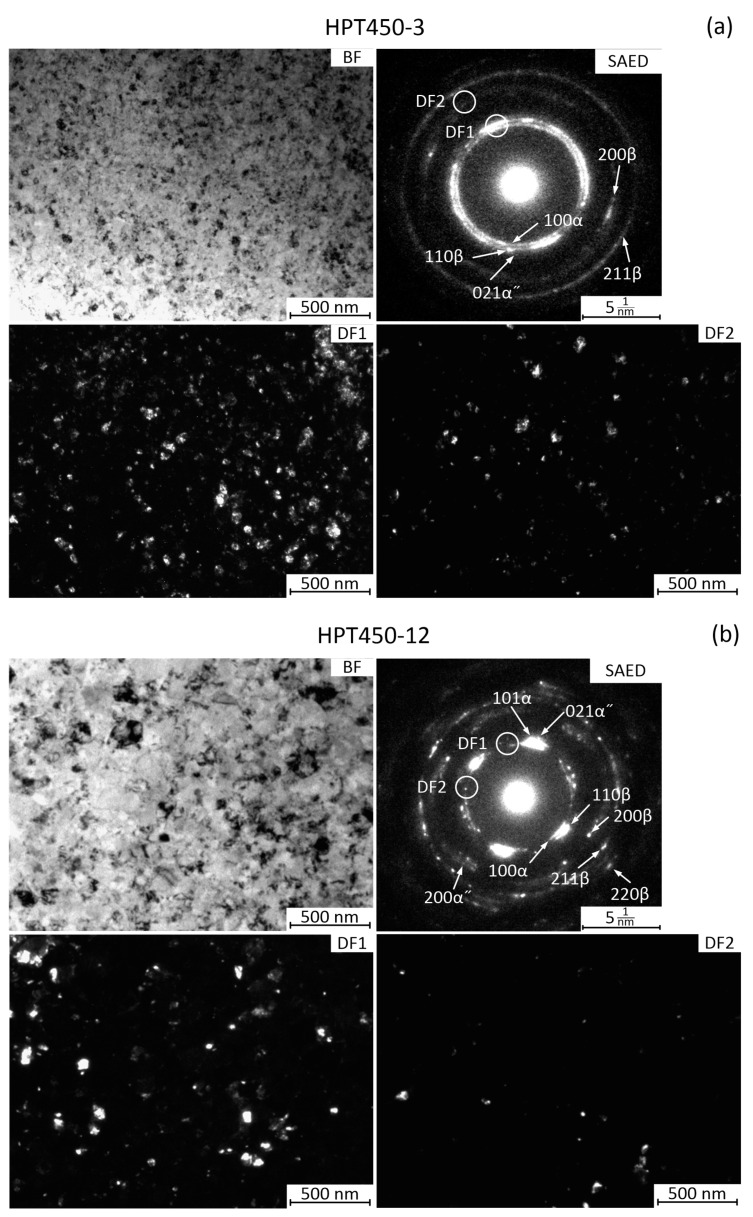
TEM images of the Ti-18Zr-15Nb alloy after: (**a**) HPT + 450 °C 3 h; (**b**) HPT + 450 °C 12 h. Bright field (BF) and dark field (DF) images, and SAED patterns. In Figure 6a, the DF1 is from both 110*β* and 100*α* rings, while DF2 is from individual 110*α* family reflex. In Figure 6b, the DF1 is taken from 100*α*, 110*β* and 021*α*″ families reflex, while DF2 is from 100*α* and 021*α*″ reflexes.

**Figure 7 materials-16-01754-f007:**
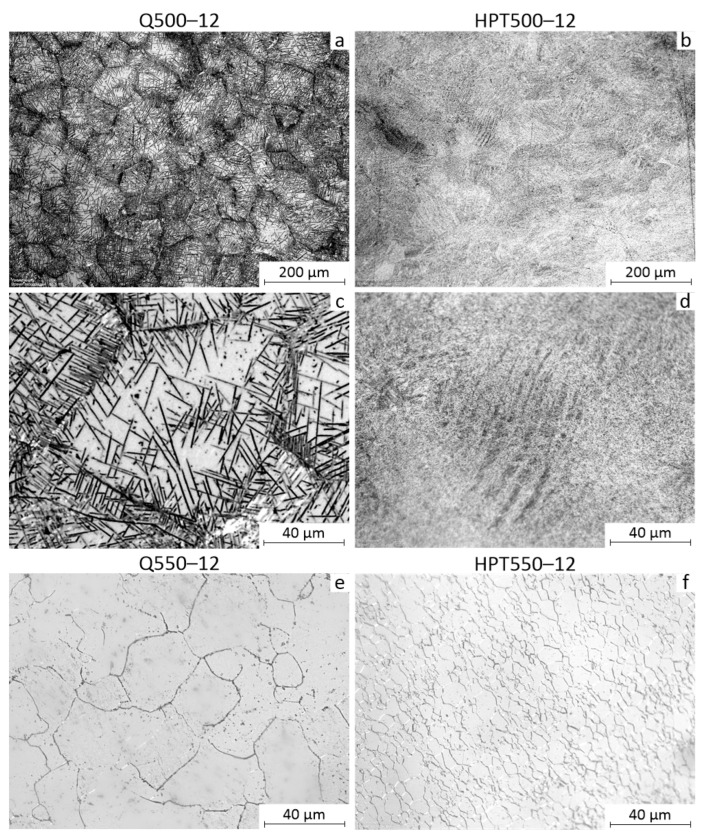
Microstructure of the Ti-18Zr-15Nb alloy after: (**a**,**c**) Q + 500 °C 12 h; (**b**,**d**) HPT + 500 °C 12 h; (**e**) Q + 550 °C 12 h; and (**f**) HPT + 550 °C 12 h.

**Figure 8 materials-16-01754-f008:**
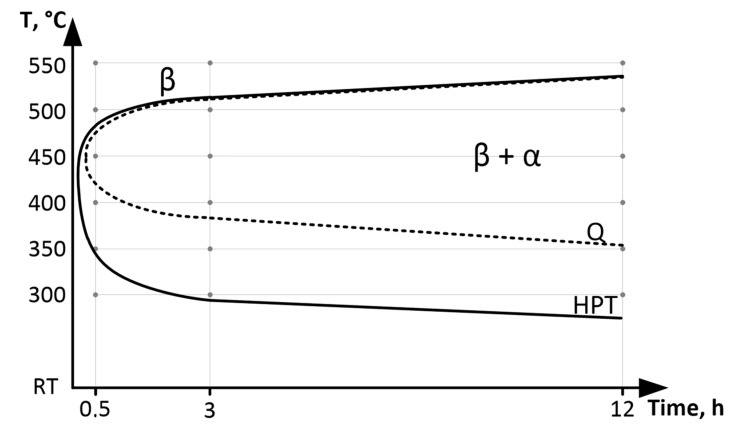
C-curve of *α*-phase formation after Q and after HPT.

**Table 1 materials-16-01754-t001:** Structure and multiplicity factors for different X-ray lines of BCC and HCP crystal structures.

Type	{*hkl*}	${\left|F\right|}_{\beta_{hkl}}^{2}$	$P_{hkl}^{\beta }$
BCC	110	$4\cdot f^{2}$	12
	200	$4\cdot f^{2}$	6
	211	$4\cdot f^{2}$	24
	220	$4\cdot f^{2}$	12
	310	$4\cdot f^{2}$	24
	222	$4\cdot f^{2}$	8
HCP	100	$f^{2}$	6
	002	$4\cdot f^{2}$	2
	101	$3\cdot f^{2}$	12
	102	$f^{2}$	12
	110	$4\cdot f^{2}$	6
	103	$3\cdot f^{2}$	12
	200	$f^{2}$	6
	112	$4\cdot f^{2}$	12
	201	$3\cdot f^{2}$	12
	004	$4\cdot f^{2}$	2
	022	$f^{2}$	12

**Table 2 materials-16-01754-t002:** Phase analysis results based on the XRD data for the Ti-18Zr-15Nb alloy after Q, HPT and aging during 0.5 h, 3 h, and 12 h at temperatures of 300–550 °C.

State	Annealing Temperature, °C	Annealing Time		
		0.5 h	3 h	12 h
Q	300	ω*	ω*	ω*
Q	350	ω*	ω*	ω*
		α phase content, %		
Q	400	-	23 ± 4	46 ± 9
Q	450	26 ± 9	29 ± 8	30 ± 4
Q	500	-	11 ± 4	13 ± 3
Q	550	-	-	-
HPT	300	-	26 **	19 **
HPT	350	15 **	25 ± 11	39 ± 27
HPT	400	35 ± 16	34 ± 17	46 ± 28
HPT	450	32 ± 12	42 ± 16	33 ± 17
HPT	500	0	34 ± 18	25 ± 15
HPT	550	-	-	-

ω*—the phase is present in a noticeable amount, but its % content is determined with a very large error. **—only one *α*-phase line.

## Data Availability

Not applicable.

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
