# Peer review of "Effect of High-Pressure Torsion and Annealing on the Structure, Phase Composition, and Microhardness of the Ti-18Zr-15Nb (at. %) Alloy"

_materials, 2023, doi:10.3390/ma16041754_

Round 1

Reviewer 1 Report

This paper deals with the microstructure and phase trasformation behavior of a new Ti-based shape memory alloy with different heat treatment conditions. The results are very interesting and the scientific quality of the presentation is high. The paper contains new results and the conclusions are well supported by the results and discussions. Therefore I would like to recommend this paper to be accepted.

Author Response

The reviewer has no comments 

Reviewer 2 Report

Dear authors,

please, find attached my comments in the PDF document.

Best regards.

Author Response

Thank you very much for the helpful comments

Comment. I would suggest incluiding in the introduction how the increase of the microhardness after HPT and aneealing procedure affects the elastic modulus and therefore this procedure and material improve the medical utilities.

Authors reply:

The corresponding clarification is added to the revised version as follows:  However, during annealing at 300-500 °C, the beta phase can be unstable and decay with the release of HCP α- and FCC w- phases [13–15]. The presence of these phases in the composition contributes to an increase in strength and deterioration of functional properties (increase in Young’s modulus and degradation in superelasticity) [13].

Comment. In the paper you refer to ECAP but do not present any results related to this technique. Do you have any experiment with this technique?

Authors reply: The corresponding clarification is added to the revised version as follows:  “The ECAP leads to formation of a predominantly nanocrystalline structure composed of equiaxed structure elements (grains and subgrains) of β-phase whose diameter ranges from 20 to 100 nm with some amount of α″- and ω-phases, and leads to an almost twofold increase in the yield strength [22,23]. The β-phase is a main phase in the Ti-18Zr-15Nb af-ter HPT with some amount of α″-phase, the grains after HPT processing are refined down to 50 nm [21].”

Comment. In line 207 you refer to table 3, however this table does not exist. I imagine you mean the second part of table 2, do not you?

Authors reply: The Table 3 is replaced by Table 2 in the revised version of the paper. 

Comment. At first glance TEM results are really interesting and the DF images are very nice. Be that as it may, in a polycristalline materials with a small grain size it is so complicated to index the diffraction spots, furthermore if you have different phases coexisting at the same time in the material, and hence, though the DF images could correspond to the phases pointed out in the paper there is not an easy way to prove it, even more so in this the case of HPT450 3 or 12 h where the grain size produces spot rings and you cannot select a single spot. As a consequence, these DF images do not provide any information, so I recommend that you must present in the paper TEM images, or alternatively

Authors reply:  The DF images are used to confirm the presence of various phases, as follows: In Fig. 5a, the DF1 is taken from the individual 211β reflex, while DF2 from 110α reflex. In Fig. 5b, the DF1 is taken from the individual 100α reflex, while DF2 from 110β reflex. In Fig. 6a, the DF1 is taken simultaneously from 110β and 100α rings, while DF2 from individual 110α reflex. In Fig. 6b, the DF1 is taken from 100α, 110β and 021α˝ reflex, while DF2 from 100α and 021α˝ reflexes. The corresponding classification are added to captions to Fig. 5 and 6.

Comment. Optical or SEM images for all annealing temperatures and times for the Q- and HPT-samples, in order to clearly see the evolution of the microstructure afte thermo-mechanical procedure. If you want to present a visual proof of the presence of the different phases you must take HRTEM images, calculate the FFT of each zone and index them to know which phases are present.

Authors reply:  The resolutions of Optical microscope or SEM allowed us to study the structure images at microscale. For the phase analysis at a nanoscale, the conventional TEM was turned out enough.

Comment. Figure 8, in my opinion, is not clear. It is necessary to explain briefly the phases present in the TTT diagram.

Authors reply:  The Figure 8 was corrected according to the reviewer’s comment

Comment. Since the Ti-18Zr-15Nb alloy is a SMA and this characteristic is key to reduce the elastic modulus and therefore enhance the medical utilities of this type of material, it would be convenient to present the evolution of the martensitic transformation temperatures with the different TM procedures as well as to present the evolution of the elastic modulus.

Authors reply:  Yes, the effect of TMT regimes and phase composition on the critical temperatures of martensitic transformations and the elastic modulus is certainly interesting and important for study. These studies will be carried out in future works.

Comment. Self-references, or references to the own group, should be reduced.

Authors reply: Some self-references (highlighted in yellow) are removed from the revised version.

Reviewer 3 Report

Dear Authors

The article is well prepared with a vast of results to support the idea about annealing sample and increase hardness due to result of equiaxed grains of needle like structure. Author need to mention the volume fraction of needle like structutre formed and the quantity of this structutre needed to improve hardness. Author also need to mention about application very well not to be general medical application. What the corrosion property and durability of this alloy in different environments?

Author Response

Thank you very much for the helpful comments

Сomment. Author need to mention the volume fraction of needle like structutre formed and the quantity of this structutre needed to improve hardness.

 Authors reply:    Evolution of the volume fraction of α-phase is based on XRD data and the results are presented in Table 2. Mention the volume fraction of needle-like structutre (α-phase) made from TEM photos for states after Q + 450 °C 3 h and Q + 450 °C 12 h has a large error, but generally agrees with the data from XRD. Evolution of the volume fraction of needle-like structutre (α-phase) from TEM photos for states after HPT + 450 °C 3 h and HPT + 450 °C 12 h is practically impossible.

Comment. Author also need to mention about application very well not to be general medical application. What the corrosion property and durability of this alloy in different environments?

Authors reply: The corresponding clarification is added to the revised version as follows:  “It was also shown in [13 Zhukova, Y.; Korobkova, A.; Dubinskiy, S.; Pustov, Y.; Konopatsky, A.; Podgorny, D.; Filonov, M.; Prokoshkin, S.; Brailovski, V. The Electrochemical and Mechanical Behavior of Bulk and Porous Superelastic Ti‒Zr-Based Alloys for Biomedical Applications. Materials 2019, 12, 2395. https://doi.org/10.3390/ma12152395], that the thermomechanically treated Ti-18Zr-15Nb alloy exhibits the sufficient corrosion resistance and high fatigue life during functional cycling in Hanks’ solution.”

Reviewer 4 Report

In the present work, the phase transformations during heating of the Ti-18Zr-15Nb alloys in the coarse-grained quenched state and the nanostructured state after high-pressure torsion (HPT) have been investigated, in order to study the stability of the β-phase during annealing under different modes of Ti-18Zr-15Nb alloy in the initial quenched state and after preliminary exposure to HPT. The results are novel and interesting. However, there are several aspects should be clarified before the publication in materials.

Major concerns:

1. As shown table 2, it is seen that the results of phase fraction analysis present a large error for the HPT specimens, may be it should be ascribed into the authors used disc specimens for the XRD characterization. In that case, the analysis of phase volume fraction according the intensity of the diffraction peak is not accurate.

2. I am not convinced on indexation of the SAED patterns for the HPT450-3 as shown Figure 6a and Figure 6b, since there is no scale bar in the SAED patterns. In addition, the α” was not detected in the XRD diffractograms of HPT 450-3 and HPT450-12, why it is appeared in the SAED patterns. The diffraction patterns for 021 α” is not clear in Figure 6a, please double check it. According to the SAED patterns in Figure 6b, the space distance of 200α” seems be smaller than the 021α”, please check it.

3. The authors state that, the α-phase at the β-grain boundaries during the annealing at 500 ℃ in the HPT state actively forms, however, in the OM images it is not clear to see the α-phase and the β phase. It is suggested to mark out the α-phase and the β phase in the OM images (Figure 7b and 7d).

Minor concerns:

Some typos should be corrected. In page11, line 300 and line 302, Figure6.a should be Figure 6a

Author Response

Thank you very much for the helpful comments

  1. As shown table 2, it is seen that the results of phase fraction analysis present a large error for the HPT specimens, may be it should be ascribed into the authors used disc specimens for the XRD characterization. In that case, the analysis of phase volume fraction according the intensity of the diffraction peak is not accurate.

Authors reply:  The corresponding clarification is added to the revised version as follows:  “It should be noted, that the main sources of large errors in the phase fraction analysis for the HPT specimens are as follows (1) the crystallographic texture formed in the HPT specimens which effect is reduced but not completely eliminated by the use of a set of X-ray lines; (2) the low accuracy of determining integrated intensities of weak X-ray lines. However, the analysis of phase volume fraction from XRD data allows understanding of the general picture of phase transformations in the HPT specimens.”

2  Comment. I am not convinced on indexation of the SAED patterns for the HPT450-3 as shown Figure 6a and Figure 6b, since there is no scale bar in the SAED patterns. In addition, the α” was not detected in the XRD diffractograms of HPT 450-3 and HPT450-12, why it is appeared in the SAED patterns. The diffraction patterns for 021 α” is not clear in Figure 6a, please double check it. According to the SAED patterns in Figure 6b, the space distance of 200α” seems be smaller than the 021α”, please check it.

Authors reply:

  • The scale bar of 5 1/nm is placed in the SAED pattern of Fig. 6a.
  • The α˝-phase X-ray lines «sink» in the X-ray diffractogram background due to its low amount. On the contrary, even weak α˝-phase reflexes can be distinctly visible due to their azimuthal and radial resolution.
  • The presence of the weak 021α˝ reflex in SAED, Fig. 6a, was checked and confirmed.
  • The correctness of indexing 200α˝ and 021α˝ was checked and confirmed. The radius-vector of 200α˝ in the reciprocal space is longer than that of 021α˝, as it should be.

The corresponding classification are added to the text. 

  1. The authors state that, the α-phase at the β-grain boundaries during the annealing at 500 ℃ in the HPT state actively forms, however, in the OM images it is not clear to see the α-phase and the β phase. It is suggested to mark out the α-phase and the β phase in the OM images (Figure 7b and 7d).

Authors reply: The corresponding clarification is added to the revised version as follows:    “The α-phase plates, about 5 μm thick and up to 50 μm long crossing the β-phase grains are observed as well (Figure 7a,c). Dark α-phase inclusions at the boundaries of β-phase grains are also clearly visible (Figure 7,c) Thus, the α-phase is actively released along the boundaries of β-phase grains during annealing, which is consistent with data from other papers [14].”

Minor concerns:  Some typos should be corrected. In page11, line 300 and line 302, Figure6.a should be Figure 6a

Authors reply: The typos were corrected

Round 2

Reviewer 2 Report

Dear authors,

I accept the paper in its presen form, but I would have liked you to explain, in detail, how the presence of both phases contributes to increase the strength of your material and how it affects the SME. Further to this, I do not question the indexing of the diffraction patterns, although I miss an explanation of the procedure used to index them. Likewise a greater diversity of authors as references would improve the article. 

In any case, I consider the paper is appropriate for the journal in its current form.

Best regards.

Reviewer 4 Report

The authors have provided a nicely detailed and thorough response to the comments from the previous review and have addressed my major concerns.  The revised manuscript is ready for publication.